# Radially distributed charging time constants at an electrode-solution interface

Ben Niu [1,4], Ruo-Chen Xie[1,4], Bin Ren [2,3], Yi-Tao Long [1] & Wei Wang [1] ✉

An electrochemically homogeneous electrode-solution interface should be understood as spatially invariant in both terms of intrinsic reactivity for the electrode side and electrical resistance mainly for the solution side. The latter remains presumably assumed in almost all cases. However, by using optical microscopy to spatially resolve the classic redox electrochemistry occurring at the whole surface of a gold macroelectrode, we discover that the electron transfer occurs always significantly sooner (by milliseconds), rather than faster in essence, at the radial coordinates closer to the electrode periphery than the very center. So is the charging process when there is no electron transfer. Based on optical measurements of the interfacial impedance, this spatially unsynchronized electron transfer is attributed to a radially non-uniform distribution of solution resistance. We accordingly manage to eliminate the heterogeneity by engineering the solution resistance distribution. The revealed spatially-dependent charging time 'constant' (to be questioned) would help paint our overall fundamental picture of electrode kinetics.

Classic electrochemistry theories are derived by assuming a homogeneous electrode-solution interface across which a potential is applied and electron transfer occurs[1]. There are two respects of meaning for the homogeneity. If the interface is considered to be electrochemically homogeneous, not only the standard rate constant at the electrode surface is the same at all coordinates, i.e., homogeneous activity, but also the electron transfer events at different locations should occur in the same pace with the electrode potential, which depends mainly on the resistance of the solution between electrodes, thus namely homogeneous solution resistance. While intensive efforts have been made to examine the activity homogeneity of various catalytically active electrodes such as the discoveries of the atomic-scale surface defects in metallic electrodes or impurities in carbon electrodes[2,3], little is known as to whether upon electrode polarization, electron transfer at different sites, even for an activity-wise homogeneous electrode, is subject indeed to a same potential. In other words, whether the cell time constant for charging an interface is indeed a 'constant' over the entire plane is a critical fundamental question. It should be more prominent in fast transient electrochemistry and is an essential aspect for our general picture of electrode kinetics.

Despite some early theoretical efforts with a focus on non-Faradaic charging process[4,5], the spatial distribution of the cell time constant has not been experimentally probed yet, likely due to the lack of characterization techniques which require both sufficient spatial and importantly, temporal resolutions. Scanning probe-based electrochemical microscopy (SPECM) enables direct mapping of local electron transfer information currently with a sub-micron resolution[6], thereby particularly powerful for revealing electrode heterogeneity in activity[7,8]. However, the speed of such imaging methods is limited by the mechanical scanning to optimally second timescale[9], insufficient for the usual timescales of cell time constant between millisecond and microsecond. More importantly, investigations on the spatial heterogeneity by SPECM would be totally affected by the use of the scanning probe because the presence of the solid probe must significantly alter the electrical properties of the solution side of the interface from the

[1]State Key Laboratory of Analytical Chemistry for Life Science, School of Chemistry and Chemical Engineering, Chemistry and Biomedicine Innovation Center (ChemBIC), Nanjing University, Nanjing 210023, China. [2]State Key Laboratory of Physical Chemistry of Solid Surfaces, Collaborative Innovation Center of Chemistry for Energy Materials (i-ChEM), Department of Chemistry, College of Chemistry and Chemical Engineering, Xiamen University, Xiamen 361005, China. [3]Innovation Laboratory for Sciences and Technologies of Energy Materials of Fujian Province (IKKEM), Xiamen 361005, China. [4]These authors contributed equally: Ben Niu, Ruo-Chen Xie. ✉e-mail: wei.wang@nju.edu.cn

original scenario. Both aspects make SPECM unsuitable to examine the time-resolved homogeneity. Another established strategy in acquiring spatial information uses localized visible light and a semiconductor electrode to confine electrochemical reactions within the projection area of light path, thereby achieving electrochemical read-out with micrometer-level spatial resolution[10,11]. Nevertheless, the light addressable electrochemistry also inevitably deviates the mass transport regime for each confined sites of an electrode from that for the original macrosurface and thus places additional obstacles to explore the question.

Unlike the established methodologies by spatially confining electrochemistry, wide-field optical electrochemical microscopy has been used to create an indirect but quantitative map on the distribution of local current density of the entire electrode surface simultaneously, according to certain optical-to-electrochemical conversion model[12,13]. For instance, in a pioneering study by Tao and coworkers[14], Faradaic reactions on an inhomogeneous electrode result in a heterogeneous distribution of refractive index, which is visualized and quantitatively processed by surface plasmon resonance microscopy to consequently convert to a map of electrochemical current density. As microscopy is a kind of wide-field imaging technique (in contrast to point-scan), the temporal resolution of capturing complete images readily reaches below milliseconds and the reaction occurs not differently from conventional electrochemical experiments. The strategy has been later on expanded to non-Faradaic process[15,16]. In addition, a variety of other optical methods have also been combined with electric potential regulation for quantitative studies of electrochemical systems, which include fluorescence[17,18], Raman[19–21], transmission[22], dark-field scattering[23–25], and reflection interference[26–29].

By integrating optical microscopy with multiple electrochemical methods, we surprisingly discover for the first time that electron transfer occurring at a well-cleaned bare electrode in a conventional three-electrode experiment exhibits significant macroscale spatial heterogeneity during dynamic potential controls. That is, under our conditions the electron transfer at the periphery region is always sooner in time than that of the central region, regardless of electrode material, scan rate and solution ionic strength. Such heterogeneity is found to exist in both cases of Faradaic reactions and non-Faradaic processes. Next, the investigations on the non-Faradaic processes using optical impedance microscopy, assisted with multiphysics modeling, enable the physical interpretation of the discovery as the radially time constant distribution of the interfacial potential establishment. Last, we manage to experimentally completely eliminate the heterogeneity which in turn reflects the validity of the proposed understanding of solution resistance.

## Results
### Imaging redox electrochemistry of gold with reflective optical microscopy

Redox voltammetry of gold has been considered as a paradigm experiment of classic electrochemistry due to its high resistivity to dissolution, very weak chemisorbing properties and an extensive double layer region in most cases[30]. The relatively large change in refractive index between gold and gold oxide and importantly the well-defined reduction peak make it ideal for studying the characterization capacity of optical microscopy. We therefore perform cyclic voltammetry (CV) of a thoroughly cleaned gold macrodisc electrode ($d = 3.0$ mm) in a 10 mM $H_2SO_4$ solution between 0.05 V and 1.50 V (vs. Ag/AgCl). Meanwhile, a monochromatic light ($\lambda = 530$ nm, corresponding to the maximum spectroscopic extinction of gold oxide as compared to metallic gold, see Supplementary Fig. 1) is directed onto the gold electrode surface and then the reflected light from the electrode surface is recorded by the camera (Fig. 1a). Note that the system is entirely the same as in the routinely performed CV experiments except the use of the non-invasive light source. Consequently, it is seen

from Fig. 1b that as the potential is scanned positively from 0.05 V to 1.05 V, the average reflected light intensity normalized to the initial intensity (i.e. reflectivity) measured from the whole electrode surface only slightly decreased from initially 100% (normalized to the maximum of the whole curve) to 98.5% (Fig. 1b). The slight decrease of reflected light is considered to result from the declined electron density and thus lowered dielectric constants at the positive potentials[31]. This also serves as a basis for the study of the non-Faradaic process later in this work.

By contrast, as the potential is further scanned from 1.05 V to 1.50 V, the optical reflectivity of the gold surface drastically decreases to 95.8%. Since the threshold potential of nearly 1.00 V corresponds well to that at which electrochemical oxidation of gold occurs via $AuOOH + 3e^- + 3H^+ = Au + 2H_2O$[32] (refer to Fig. 1c for the voltammetry) and given the fact that the formed gold oxide is less reflective than metallic gold, the reflectivity change indicates the visualization of gold oxidation. Little change is observed at the reversed potentials between 1.50 V and 1.05 V, suggesting the electrochemical stability of gold oxide. When the potential further reverses negatively from 1.05 V to 0.70 V, the optical intensity of the electrode surface more drastically increase back to 98.4%, corresponding to the reduction of the formed gold oxide. The faster process as compared to the oxidation reaction is the focus of our following study as mentioned above.

Overall, the optical reflectivity variation is plotted as a function of time and is seen to repeatedly and reversibly vary in pace with the potential ramps applied. Since the change in optical intensity has been well shown to be proportional to the quantity of charge transferred from the electrode[13,24], differentiating the intensity over time gives the optical current. As such, further by converting the timescale to the potential scale, we obtain the optical voltammogram (orange curve in Fig. 1c) that consequently exhibits nearly perfect consistency particularly in terms of the potential of the sharp reduction peak at 0.801 V with the electrochemical CV (i.e. 0.805 V, blue curve). This evidences the quantitative ability of the optical microscopy in measuring electrode kinetics.

### Mapping the distribution of apparent electron transfer kinetics

The advantageous pixel-scale resolution of optical microscopy allows extraction of the optical CV from every pixel of the electrode image. The experimental difficulty in acquiring high signal-to-noise ratio data lies in the dominant short-term noise from the light source, especially for the derivative of intensity as it would amplify the noise. Accordingly, this type of noise is removed by introducing an optical reference of a silicon wafer placed in the same view (Supplementary Fig. 2). We thus compare the potential-determining reflectivity from the center region with that from the edge areas. The variation of reflectivity in the edge region initiates and completes earlier than that in the center region (Fig. 2a). Evident from the converted optical CV, the peak potential of the edge area is 24 mV more positive than that of the center area (Fig. 2b). Note that Gaussian fit is conducted across the peak current to accurately extract the peak potential. A full pixel-resolved mapping of the peak potentials is then obtained showing a gradual and radially symmetric variation (Fig. 2c, d). At this point, we have performed tilted scanning electron microscopic measurements, showing highly flat electrode geometry over the entire surface without any marked curvature (Supplementary Fig. 3). Furthermore, systematic results in Supplementary Figs. 4, 6 confirm that the same pattern is always present despite of the applied scan rates ($0.1\,V\,s^{-1} \sim 2\,V\,s^{-1}$), solution ionic strength (2 mM–0.5 M). In particular, we have examined both oxidation and reduction processes of Prussian blue electrochemistry and observed the same rules as well (Supplementary Fig. 7). These results confirm the experimental validity of the observed peak potential shift with spatial coordinates.

Given the millimeter scale of the continuous spatial heterogeneity, we exclude the possibility of the well-known 'edge effects' by radial (as

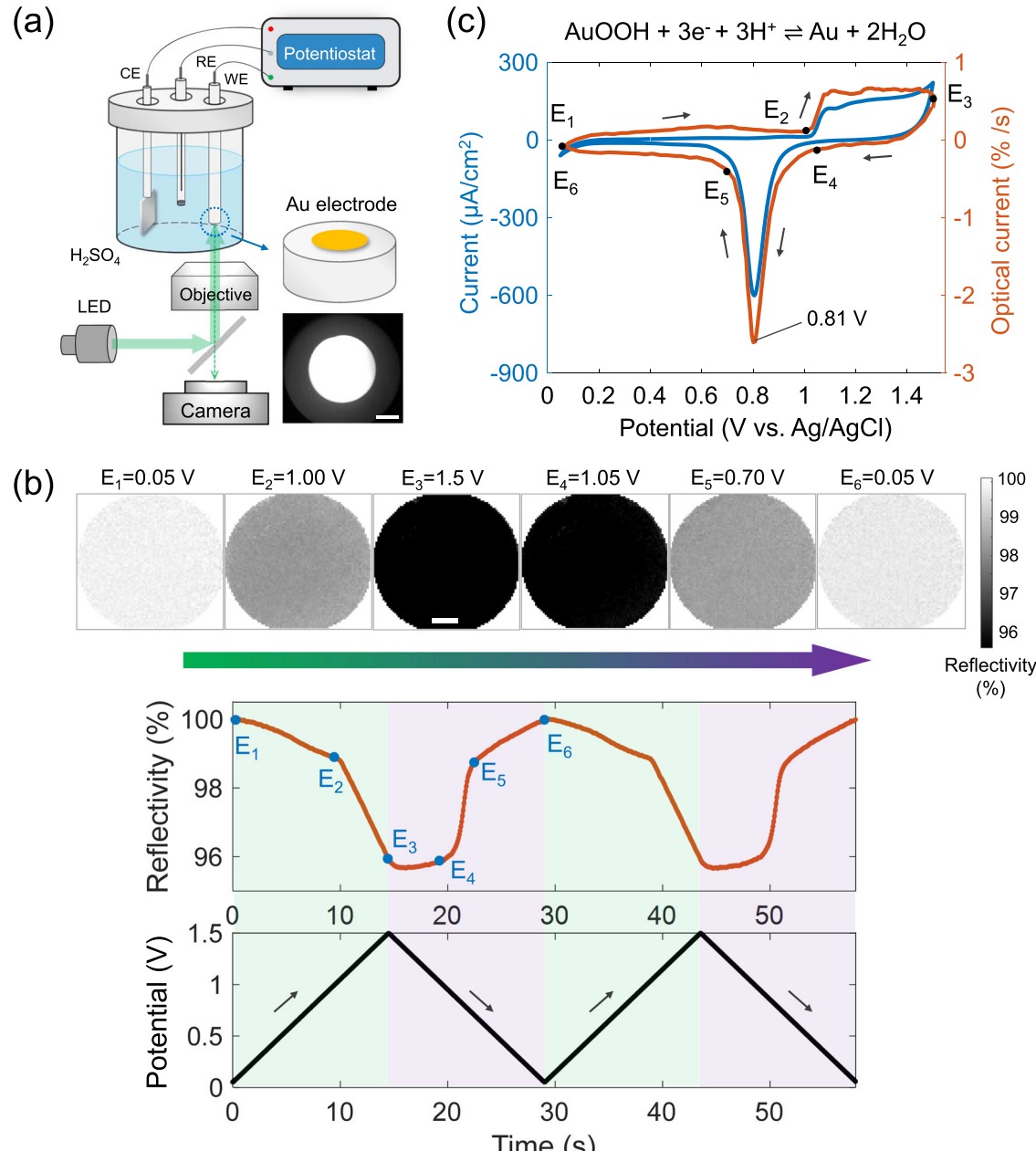

**Fig. 1 | Imaging redox electrochemistry of a gold electrode with reflective optical microscopy. a** Reflective optical imaging of a gold electrode surface during the classic voltammetric experiment using the three-electrode system. **b** Optical images (top charts) and mean reflectivity (middle plot) of the overall electrode surface as a function of applied potential (bottom plot). Scale bar 0.5 mm. **c** Cyclic voltammetry of the gold electrode (blue) overlaid with derived optical current (orange). Scan rate 100 mV s⁻¹. Solution: 10 mM $H_2SO_4$. Temperature: 20 °C. Source data are provided as a Source Data file.

opposed to linear) diffusion towards the periphery of a disc electrode[33], as reported by Pan et al.[19] where the variation of the measured peak potential distributions of methylviologen is found only at the edge region with a lengthscale of ~100 μm at a timescale of 10 s, consistent with diffusional behavior. Radial diffusion is also simulated in our scenario to show its significant occurrence constantly and only at very confined space near the electrode edges (Supplementary Fig. 8). Consequently, the measured peak potential difference between central and edge regions indicates a spatial difference of electron transfer kinetics. It follows that either the standard electrochemical rate constant $k_O$ differs from one site to another, or it has nothing to do with the heterogeneous kinetics – merely the potentials to be imposed for reduction is actually reached with a difference in the time delay, which would be directly related to the process of solution phase charge transport.

## Mapping the distribution of charging time constants without electron transfer

We suspect the latter possibility for two reasons. First, a columbic estimate of reduction from the electrical CV gives a value exactly consistent with that from the expected monolayer reaction for gold (see Supplementary Note 8 for detailed discussion). This implies that the monolayer reaction is unlikely to exhibit significant activity heterogeneity as compared to many redox reactions involving solid state diffusion and phase transition. This also reflects our careful selection of gold for this work. Second, the solution ionic strength is observed to largely affect the magnitude of the heterogeneity. This suggests the possible role of solution phase charge transport. We thus remove the process of electron transfer and look only into the non-Faradaic electrode processes.

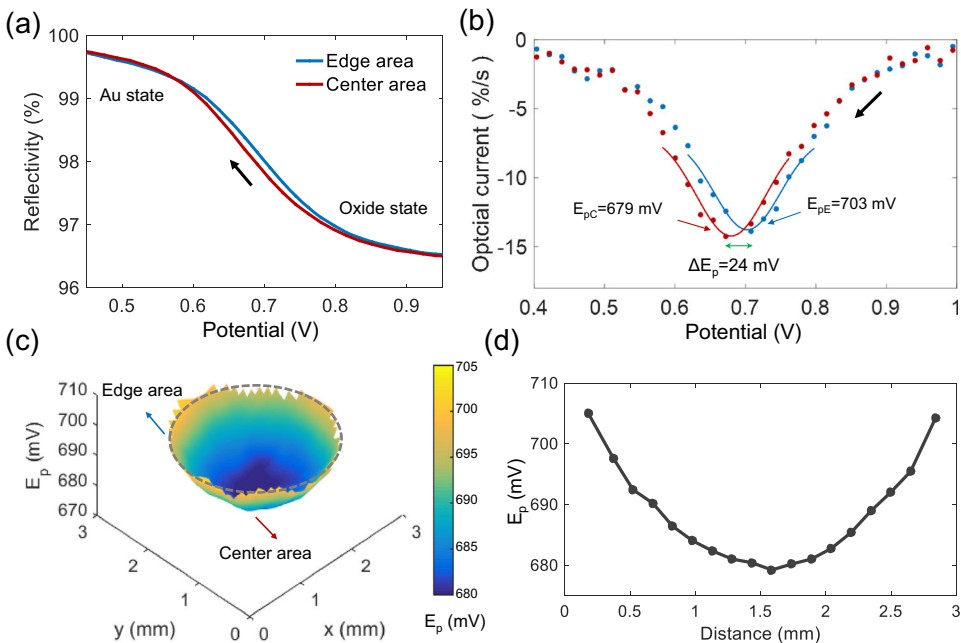

**Fig. 2 | Mapping the distribution of reduction peak potentials over the electrode surface. a** Potential dependent optical reflectivity measured for the edge and center areas respectively. Scan rate 1 V s⁻¹. **b** Reductive optical voltammetric peaks of the edge and central electrode areas. The line is the fitted results, and the dots are experimental data. **c** Pixel-level peak potential distribution over the whole electrode surface and **d** the representative radial peak potential distribution. Source data are provided as a Source Data file.

Experimentally, a potential step is applied from 0.6 V to 0.2 V where only charging of the electrode occurs. Optical reflectivity of the central and the edge regions is recorded and compared (Supplementary Fig. 10). The variation in the reflectivity in Fig. 3a shows that the dynamic process overall takes nearly 20 ms, consistent with the generally milliseconds' cell time constants (kΩ×μF) of charging a macroelectrode.

Most importantly, the temporally evolving reflectivity exhibits discrepancy between the same two kinds of regions and the rise in the reflectivity of the edge area is clearly sooner. The optical images at different times are presented in Fig. 3b. It is observed that as the potential is initially held at 0.6 V, the electrode surface appears completely homogeneously bright, but upon the application of the potential step, the pattern with a darker center is visualized and sustains almost throughout the 20 ms of the charging before becoming generally uniform again. Figure 3c summarizes the planar distribution of the extracted time constant from the reflectivity curves, showing a gradual, continuous increase at the millimeter lengthscale from the periphery region continuously towards the center of the electrode surface. Possible effects of uneven illumination and focal plane on the optical measurements have been carefully examined and confirmed to be negligible (Supplementary Note 10). Given the extremely fast electron transport in the gold phase (2 m² V⁻¹ s⁻¹)[34] as compared to the ionic mobility in the solution phase (10⁻¹⁰ -10⁻⁷ m² V⁻¹ s⁻¹)[35], the mapping results of the non-Faradaic charging indicate that the charge transport in the solution side of the interface is not of a same rate across different locations.

## Imaging the distribution of electrochemical impedance

To unravel the physical basis of the discrepancy in the interfacial kinetics, the following study of the charging process at the electrode-solution interface is then based on the analysis of electrochemical impedance. Experimentally, we conduct optical electrochemical impedance microscopy (oEIS) to deconvolute the respective effects of the double layer capacitance ($C_{dl}$) and solution resistance ($R_s$). Detailed descriptions are provided in Supplementary Note 11. Briefly, a

sinusoidal potential modulation with an amplitude of 100 mV is applied at a range of frequencies (from 0.1 Hz to 200 Hz) and consequently the optical reflectivity variation as a function of potential modulation is measured (Supplementary Fig. 15a). Then, the frequency-dependent optical amplitudes are extracted using Fourier transform (Supplementary Fig. 15d) and converted to the relative scale of the overall impedance which is then plotted as a function of the modulation frequency[15,24] (Fig. 4a). Consequently, the experimentally measured impedance ($Z$) is found to be well consistent with classic RC circuit[36], which consists of $C_{dl}$ and $R_s$ in series, following the equation:

$$Z = R_s + \frac{1}{j\omega C_{dl}} \qquad (1)$$

where $\omega$ (=2π$f$) is the angular frequency and $j$ is the imaginary unit ($j^2 = -1$).

Building on the deconvoluted electrical components of the gold-solution interface, we now extract the planar distribution of the respective components at characteristic frequencies. At low frequency of 0.1 Hz, the impedance modulus approaches the constant value of 1/($\omega C_{dl}$) and thus Fig. 4b shows the mapping of $C_{dl}$ that is found to remain basically unchanged from one site to another. This is anticipated since the thickness of the double layer is at merely the nanometer scale which is unlikely to be of a spatial heterogeneity continuously across distances of millimeters. By contrast, as dominating (i.e. 98% at 100 Hz, Supplementary Fig. 15b) in the high-frequency domain, the solution resistance is extracted and mapped. Consequently, Fig. 4c shows the mapping that exhibits a pattern of a maximum of solution resistance in the center whereas a minimum in the periphery region. This is exactly consistent with the hump-like radial distribution discovered above in the potential step measurements where a central maximum of time constant is observed. The same experiments have also been performed using a chemically inert glassy carbon electrode and the distribution of a very similar pattern is observed (Supplementary Fig. 18), demonstrating the independence of the occurrence of the heterogeneous charging phenomenon on the

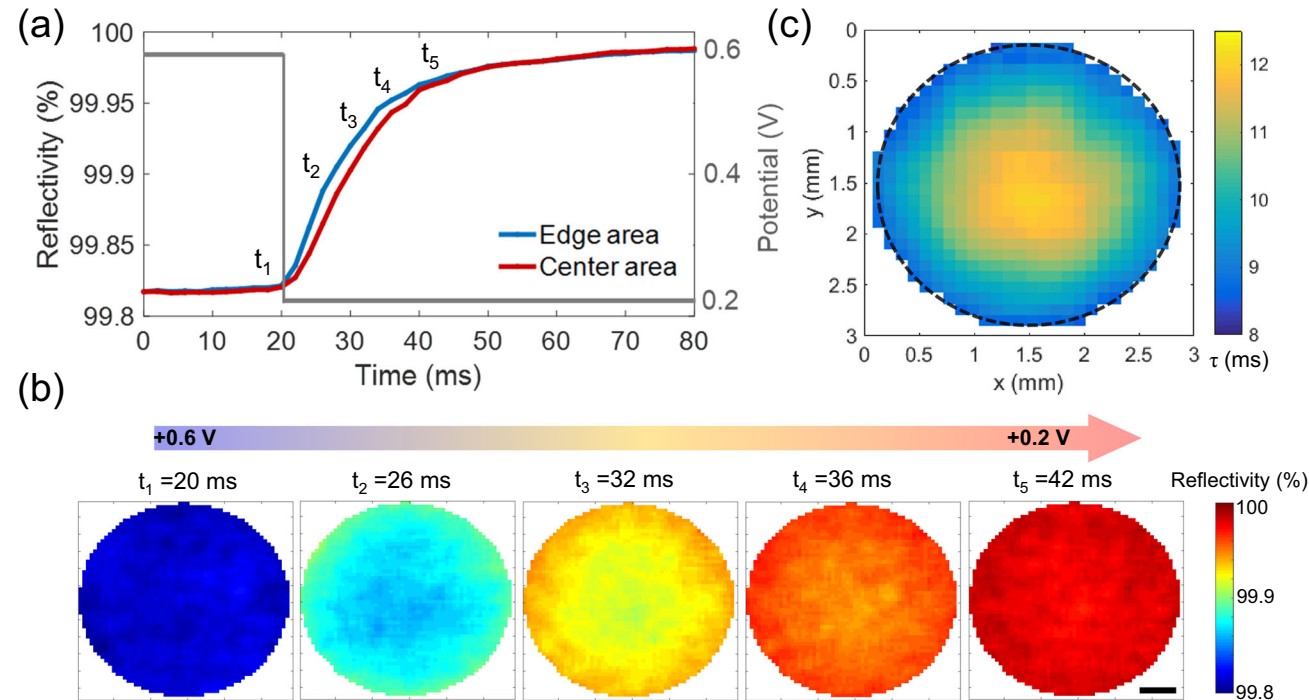

**Fig. 3 | Mapping the distribution of charging time constants of the gold surface.** **a** Potential step and corresponding reflectivity of the gold electrode at the edge and center areas. **b** Corresponding optical reflectivity images at the different charging times. Scale bar 0.5 mm. **c** Planar mapping of the extracted time constant from the optical curve. Source data are provided as a Source Data file.

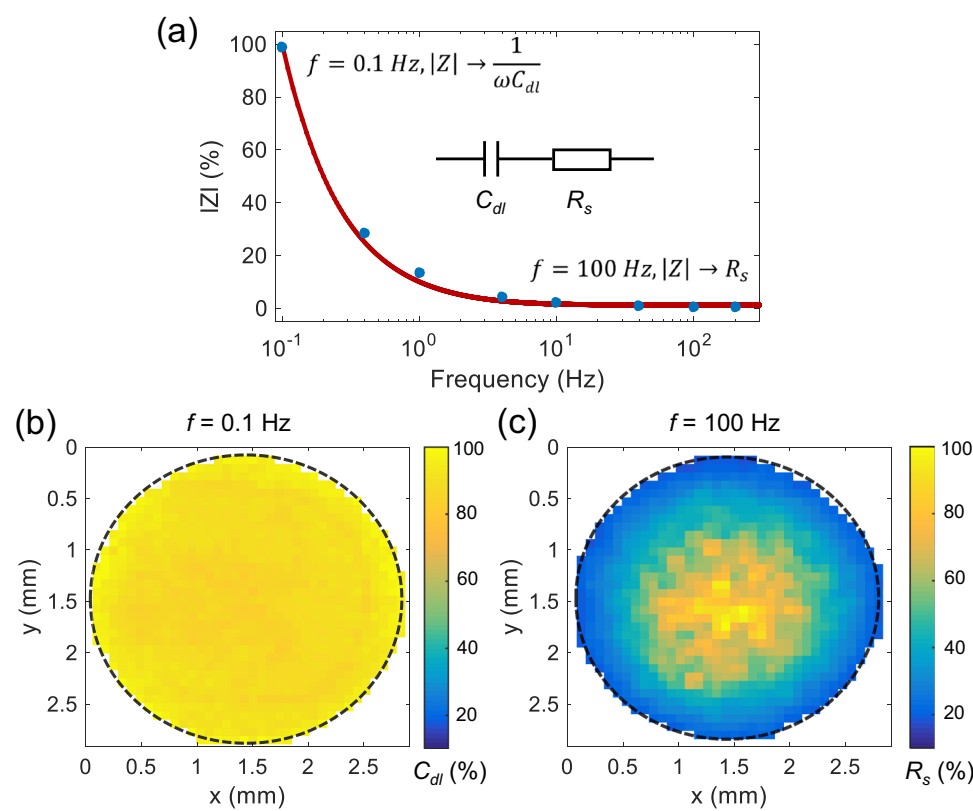

**Fig. 4 | Mapping the distribution of electrochemical impedance of the charging process.** **a** Calculated impedance modulus as a function of potential modulation frequency. The line is the fitted results, and the dots are experimental data. Planar distribution of impedance modulus at (**b**) low frequency of 0.1 Hz and (**c**) at high frequency of 100 Hz over the electrode surface. Source data are provided as a Source Data file.

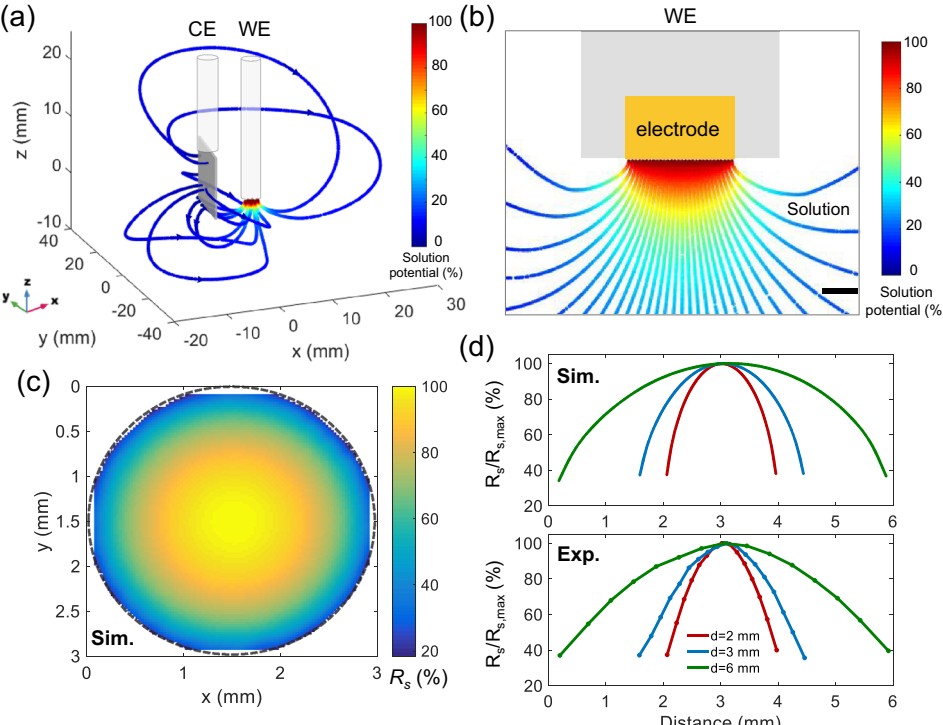

**Fig. 5 | Theoretical modeling of the $R_s$ distribution. a** COMSOL simulation of electric field lines between a working electrode and a counter electrode and (**b**) a closer look at the profile of potential distribution from the nearby solution to the working electrode surface. Scale bar 0.5 mm. **c** Top view of the solution resistance converted from the electric field strength over the entire electrode. **d** Dependence of the solution resistance distribution on the electrode diameter from simulated results (top panel, labeled as Sim.) in comparison with that measured experimentally (bottom panel, labeled as Exp.). Source data are provided as a Source Data file.

electrode material. As such, these results strongly indicate that it is the differences in the solution resistance between the central and periphery regions which cause the spatial heterogeneity of the measured kinetics of the charging process.

## Theoretical modeling of $R_s$ distribution

We also resort to COMSOL modeling for understanding the phenomenon in terms of the potential establishment starting from the working electrode all the way to the current-passing counter electrode. Figure 5a depicts the three-dimensional profiles of the electric field lines in the solution between each sites of a working electrode and a counter electrode. The distribution of the electric field does exhibit spatial heterogeneity. Interestingly, the variation is observed to be monotonic from the center to the edge along the radial direction, rather than in the intuitive far-or-close direction between the counter electrode and the working electrode.

Consequently, the overall picture of the solution phase potential distribution shows that the Ohmic drop in the solution mostly (~90%) concentrates on the proximity of the electrode surface within a distance of nearly 3 mm (Fig. 5b). Meanwhile, the stronger electric field distribution at the near-electrode space, particularly at the electrode periphery, results in significantly faster ion mobility and interfacial potential establishment (Supplementary Fig. 19). Chemically, the phenomenon is also consistent with our perception that ions tend to migrate slower as the space becomes crowded for them to move, especially considering the electrical repulsion between the molecules of same-signed charges. As such, calculating the reciprocal of the electric field strength provides the total solution resistance for each electrode site which are overall depicted for the whole interface. Detailed descriptions are provided in Supplementary Note 13. The planar and radial distributions of the solution resistance over the electrode are seen in Fig. 5c and d respectively, exhibiting the similar

hump-like shape. These patterns are exactly consistent with those experimentally measured as presented above.

Common practice suggest that a parallel two-electrode configuration may exhibit most uniform electric field. However, it is reminded that classic electrochemistry requires the counter electrode to be far larger (at least by one order of magnitude) than the working electrode. Simulation of such an electrode configuration shows that there still exists great spatial heterogeneity of the near-electrode potentials, though to a significantly lesser extent (but still marked in magnitude) when the two electrodes are placed closer (Supplementary Fig. 20). Further, as the diameter of the electrode scales between 2 mm and 6 mm, there is always present the hump-like distribution and the solution resistance over the very central region is constantly *ca.* 60% larger than that over the edge regions. These are also consistent with the trend observed in the experimentally determined variations of the solution resistance for the differently sized electrodes using the optical electrochemical impedance microscopy. Interestingly, as the gold surface is passivated along one disc diameter creating additional 'edges', the mapped solution resistances, both experimentally and theoretically, exhibit an altered discrete, bimodal distribution (Supplementary Figs. 21, 22). The results support the revealed key role of electric-field-determining solution resistance.

The origin of the spatial heterogeneity of the charging times has now been demonstrated to result from the unsynchronized charge transport in the solution phase due to the uneven electric field strength distribution and thus the radially varying solution resistance over the plane of the electrode surface. In extreme cases, greatly differing solution resistances between a tip-substrate gap and the taper-substrate zones can be used to electrodeposit precise microstructures via differently delayed redox electrochemistry[37]. It is thus reasonable to infer that the redox electron transfer of the gold surface occurs

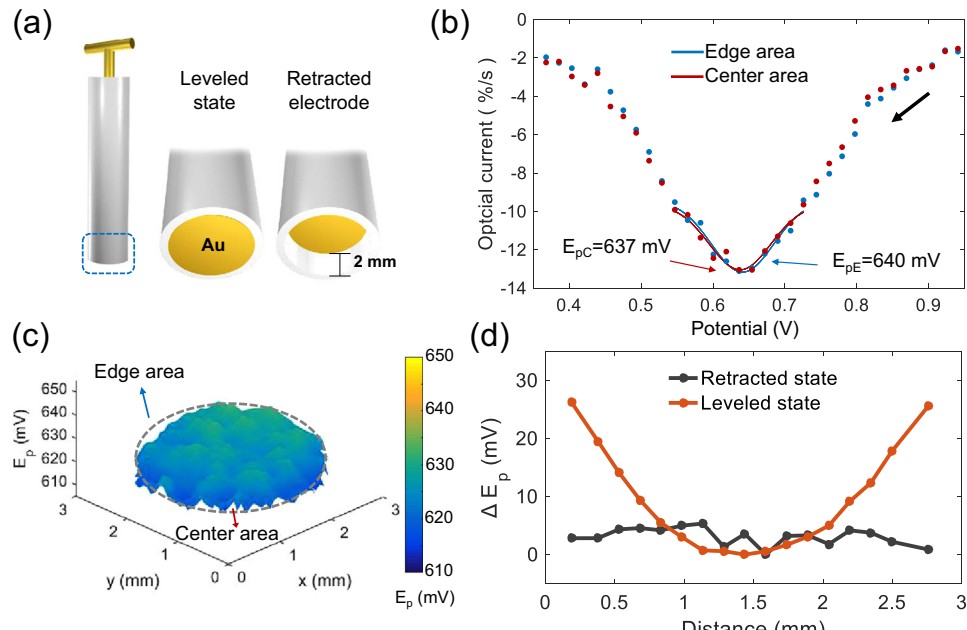

**Fig. 6 | Spatial distribution of the reduction peak potentials of a retracted electrode. a** Illustration of the retracted electrode including two different state. **b** Reductive optical voltammetric peaks of the periphery and the center areas of the retracted gold electrode. The line is the fitted results, and the dots are experimental data. **c** Optical mapping of the peak potentials across the overall electrode surface. **d** Comparison between the radial distributions of the peak potential for the leveled electrode and for the retracted electrode. Source data are provided as a Source Data file.

likewise earlier, rather than faster in essence, in the sites closer to the electrode periphery than the central region.

## Spatial distribution of redox voltammetry of a retracted electrode

To further validate our understanding above, we last seek to experimentally eliminate the heterogeneity based on the electric field distribution understanding. It is reminded that the depicted variation of solution potential is predominantly confined within the near-electrode space for all the electrode surface sites. This space corresponds to the part of the ion transport routes that become spatially compacted. It follows that the solution resistivity probably results from how accessible a location on the electrode is for ions to move between itself and the vast bulk solution. To equalize the accessibility of the electrode surface, one solution is to place the conductive surface of the electrode inwards such that no differences in the solution space above and near the electrode surface would exist amongst different locations.

Experimentally, the gold surface is retracted from the leveled plane with the resin sheath surface to 2 mm inwards (Fig. 6a), creating a cylindrical channel to confine the electrode nearby region within this space. Optical voltammetry of the retracted electrode surface is performed again but yet in this scenario, the previously seen shift of the peak potential now shows 0.637 V in the central area versus 0.640 V for edge areas (Fig. 6b), almost vanishing. The elimination of the spatial differences in the peak potentials is evident in the planar mapping (Fig. 6c) and the radial distribution (Fig. 6d). Also, the results are supported by the simulated results of the solution potential distribution where the potential drop near the electrode surface is now completely confined within the created channel, such that no radial differences in the solution resistance are present anymore (Supplementary Fig. 23). The anticipated outcomes using the retracted electrode strongly confirm our proposed physical understanding for the simulated evolution of solution potentials that the near-electrode solution is differently resistive for different electrode locations due to the varying electrode accessibility, thereby resulting in a spatial distribution of solution resistance over the electrode surface.

## Discussion

We have experimentally discovered by applying optical microscopy with cyclic or potential step voltammetry that the apparent kinetics of both Faradaic process and non-Faradaic charging process have characteristic (not random) spatial variation on a bare, well-polished gold macrodisc electrode. By using optical electrochemical impedance microscopy, we deconvolute out the factor of charge transport rate at high frequency domain of potential modulation. It is concluded that the origin of this variation lies in the non-uniformly distributed solution resistance over the electrode surface. Results from simulation and the experiment using a retracted electrode further confirm this understanding. As a universal property of an electrode-solution interface that would contribute to our general knowledge of electrode kinetics, this previously largely unexplored fundamental may be critical for fields including design of electrochemical instrumentation, ion batteries, electroanalysis of heterogeneous electrode catalysis, electrodeposition of high quality films and fast transient electrochemistry where both short-time behavior and spatial uniformity are important. In particular, our work leads to an important caveat in electroanalysis that catalysis on a spatially heterogeneous electrode can be significantly affected by, apart from the intrinsic activity of the electrode surface, also the solution resistance.

## Methods

### Cleaning of working electrode

The working electrode is a gold disc electrode of 3 mm in diameter embedded in a Teflon cylinder (99.99% purity, Tianjin Aida Hengsheng Technology Development). To maximize the surface uniformity and cleanness, the electrode is polished with different grades (1 μm, 0.3 μm, 0.05 μm in sequence) of alumina powder on a polishing cloth, followed by 1 min sonication in deionized water (DIW, 18 MΩ·cm, Milli-Q, Thermo Fisher) and then dried with blowing nitrogen gas. Next, electrochemical cycling is applied to the polished electrode in a fresh prepared 10 mM sulfuric acid (AR grade, 98%, Sinopharm Chemical Reagent Co., Ltd.) solution at the potentials between 0.05 V and 1.50 V

(vs. Ag/AgCl) with a scan rate of $0.1\,V \cdot s^{-1}$ for nearly 20 cycles until the reduction peak current is approximately a constant value. The reference electrode was calibrated prior to electrochemical measurements by immersing it in a standard solution of saturated KCl and ensuring that the recorded potential corresponded to the theoretical value at room temperature.

## Electrochemical system and reflective optical configurations

A three electrode system is used in this work ensuring precise potential controls using an Autolab potentiostat (PGSTAT302N). Apart from the gold working electrode, the reference electrode is an Ag/AgCl electrode (saturated KCl), and a platinum plate ($1\,cm \times 1\,cm \times 0.1\,mm$) serves as the counter electrode. The electrodes are assembled in a glass beaker of height 50 mm and diameter 40 mm through which light is illuminated in the way as described below. All the electrochemical measurements are conducted at room temperature of approximately 20 °C.

The imaging system was performed on an inverted microscope (Eclipse Ti-U, Nikon), with a 530 nm light-emitting diode (FWHM = 33 nm for the central wavelength, M530L3-C5, Thorlabs) as the light source throughout the work. The light was collimated and illuminated directly upwards onto the electrode through a 4X objective (N.A. = 0.13). The back-reflected light was collected by a CCD camera (Stingray, Allied Vision Technologies).

## Simultaneous optical and electrochemical voltammetry

Cyclic voltammetry of the gold electrode was performed in a 10 mM sulfuric acid solution within a volume of 20 mL between 0.05 V and 1.5 V (vs Ag/AgCl) at the scan rates ranging from $0.1\,V \cdot s^{-1}$ to $2\,V \cdot s^{-1}$. The current as a function of potential is recorded using the potentiostat. Meanwhile, the reflected light from the gold surface is measured at a frame rate of 100 in pace with the applied potentials. Synchronization between the electrical and optical signal is achieved with a data acquisition card (USB-6250, National Instruments) to allow real-time mapping.

## Optical electrochemical impedance imaging

For potential step voltammetry, a step of −0.4 V is applied from 0.6 V to 0.2 V, whereas for electrochemical impedance measurements, a sinusoidal modulation with an amplitude of 100 mV (offset 400 mV) is applied over a frequency range of 0.1 Hz to 200 Hz. The potential control is achieved by the Autolab potentiostat and modulated via an external waveform function generator (RIGOL, DG1000Z). Time-lapsed reflected images are captured at a constant speed up to 550 frames per second. The optical amplitude is determined using Fast Fourier transform analysis in MATLAB, where it is defined as half of the peak-to-peak value in the reflectivity curve.

## Theoretical modeling

Physical modeling of solution resistance distribution is based on COMSOL Multiphysics software. The simulation employs a 3-D axisymmetric geometry. The working electrode is a 3 mm diameter disk embedded in a Teflon cylinder measuring 6 mm in diameter. The counter electrode takes the form of a cuboid with dimensions of $1\,cm \times 1\,cm \times 0.1\,mm$, positioned 9 mm away from the working electrode. Appropriate boundary conditions are then applied to WE and CE, while the remaining domain of the simulation represents the solution. The mesh division is adjusted to be more refined in close proximity to the electrode surface to achieve a higher level of accuracy.

## Data availability

The data that support the findings of this study are provided in the Source data file. Source data are provided with this paper.

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

## Acknowledgements

We thank the National Natural Science Foundation of China (Grants 21925403, W.W.), the Natural Science Foundation of Jiangsu Province (BK20220768, R.-C. Xie) and the Excellent Research Program of Nanjing University (Grant ZY JH004, W.W.) for the financial support.

## Author contributions

B.N. designed and carried out the experiments and simulations. B.N., R.-C.X., and W.W. analyzed the data, discussed the results and wrote the manuscript. B.R. and Y.-T.L. helped discuss the results. W.W. conceived and supervised the research.

## Competing interests

The authors declare no competing interests.
