## [Peer Review File · Nature Communications]

REVIEWER COMMENTS

Reviewer #1 (Remarks to the Author):

In this manuscript, Dr. Wang and colleagues reveal that electron transfer occurs significantly faster at the periphery of the electrode than in the central region during redox reactions, as observed through optical microscopy. By integrating multiple electrochemical techniques with electrochemical impedance microscopy and COMSOL simulations, this study spatially correlates this variation to the non-uniform distribution of solution resistance across the electrode surface. This is a very interesting phenomenon that could be significant for electrochemical field which could impact the catalyst design research and industry. Given the importance of this paper, I strongly recommend to publish the paper in Nature Communications with minor revisions.

1. In this manuscript, the authors provide the electrolyte as the 0.01M H₂SO₄ solution. Could the authors explain the reasons for this option? Will the spatial difference still be observed on the electrode surface during the redox reaction if the concentration increases to 1 mol/L?
2. In Fig. 1b, the authors choose the different points and share the optical image during the CV scanning process. Specifically, we are interested in knowing why at some points such as E2 and E5 are selected as inflection points while E3 is at 1.5 V, which is not the inflection point of reflectivity. Additionally, we would like to know if the delays between the potential and reflectivity, like E3, are common in the data and why they occur.
3. Even though the authors provide the COMSOL simulation results of the electricity field distribution as the working electrode (WE) did not face the counter electrode (CE), as the practical experiments knowledge the WE facing the CE could receive the best electricity field. In this case, could the authors provide some simulation results with the WE and CE face-to-face each other?
4. Can you explain the reason for this uneven resistance distribution in more detail? In Fig. 5 and 6, the author seems to have changed the distribution of the solution potential on the surface through physical means. In this case, will the electric field distribution be the same as the electrode is totally in contact with the solution? Authors could artificially establish an insulating boundary along the diameter of the gold electrode surface and verify if there is a difference in response only at the edge and other areas. In addition, we are also curious about the difference in solution impedance under this uneven distribution. Is there any experimental method to quantify this difference?
5. In addition, there are some minor issues that need to be corrected:
 - ☐ If the figure abbreviations in Fig. 1a are not marked in the text, they need to be used in full.
 - ☐ Vs in Fig. 1b should be vs.
 - ☐ In line 133, Fig. 1d missed.

☐ In line 169, the constant of k_0 is not provided with the meaning and explanation.

☐ The "space" distance in Supplementary Fig. 5a and Fig. 4a seems inappropriate, please check whether there is a problem of mixing full-width and half-width.

Reviewer #2 (Remarks to the Author):

The manuscript reports an important experimental discovery of macroscale spatial heterogeneity on the electrode surface during dynamic electrochemical potential scan, for both faradaic reaction and non-Faradaic processes. Taking advantage of wide field reflective optical imaging method developed by the lab previously that can map local electrochemical current and impedance, the spatial heterogeneity is found to be caused by the solution phase resistance variations near the electrode surface, likely contributed from the ionic crowding effect. The study is well designed with proper controls to rule out other possibilities and a theoretical modeling to explain the principle of the observed phenomena. Furthermore, a simple experimental validation with inclined electrode showing that the heterogeneity can be removed completely. The conclusion is fully supported by the data presented. This finding contributes to the fundamental knowledge of electrochemistry kinetics and could have broader impact on the study and application of fast transient electrochemistry. I believe the manuscript is worth publishing in *Nature Communications* given the potential broader impact of the findings. I only have the following minor comments:

Page 10, line 274, should "smaller" be "larger"? Figure 5 C and D showing the center area solution resistance is larger.

Minor English issue:

The English of the article could be further polished. One mistake in multiple place is using "which" without putting a comma before it.

Page 3, line78, a comma should be placed before "which".

Page 5, line 121, “corresponds well to that at which electrochemical oxidation of gold occurs” is better be “corresponds to the gold electrochemical oxidization potential”.

Page 12, line 337, “which” should be “, which”.

Reviewer #3 (Remarks to the Author):

In this work Niu et al., use opto-electrochemical methods to explore heterogeneities in reactivity at seemingly homogeneous Au microspheres. Using bright-field optical microscopy the authors infer that edges of the electrode react sooner than the periphery, due to the solution resistance (rather than the electrode itself) being inhomogeneous. They then use their results to create homogeneous solution resistances and hence remove heterogeneities. They support their results with COMSOL simulations. While the work has merit in parts, I think it is seriously under-developed and not suitable for publication in Nature Communications (or any other journal for that matter at this stage). Once the below points have been addressed, I believe it could be suitable for a specialist physical or electrochemistry chemistry journal. Below I list major and minor issues with the work:

1. There has been a huge interest in using optical methods to explore electrochemical processes due to its ‘high’ read-out rate (as compared to scanning probe techniques) and non-invasiveness, as the authors highlight. However, many works combining optics and electrochemistry, (often in high-impact journals) contain very poor optical controls leading to imaging artifacts being potentially interpreted as new (unreproducible) physics. These issues I fear may also plague this work, and I have the following questions:

a. How homogeneous is the illumination of the electrode? A drawback of bright field microscopies is typically across the field of view illumination is not homogeneous, this can mean imaging at edges is challenging even if data is normalised? How do the authors tackle this issue?

b. The authors have cropped(?) their images into circularly illumination regions – potentially due to the above problem? But this then draws into question what is the edge of the electrode? How curved is the electrode they are working with? For a low NA (0.13) objective how does the depth of field of the objective influence what they are observing? Could some tilted SEM done to at least get an idea of the electrode topography?

c. The authors are imaging a complex refractive index system i.e., through water/at a metallic interface. The imaging plane of the edges could be offset from that of the centre due to imaging

through the medium. I would have expected some optical modelling e.g., in Lumerical to characterise this. Such an effect will become worse when moving to higher NAs and resolution.

d. The correlation between reflectivity and electrochemistry at the crux of the paper is poor. Factors such as how potential influences the microscope focus position (as demonstrated in the work of the Faez group e.g., Electric-double-layer-modulation microscopy *Physical Review Applied* 13 (4), 044065) are not considered. This is not a trivial effect that must be discussed.

e. Optical imaging at interfaces is generally tricky. This is because at this point there are large refractive index differences e.g., between the solution and the electrode. This results in chromatic aberrations where light travels a different path length through the medium resulting in enhanced, but artificial, reflection contrast at edges. This was shown recently for imaging in battery materials by Pandya et al. *Nature Nanotechnology* 18, 1185–1194 (2023). How do the authors overcome this? Why do they chose 530 nm as a wavelength? What happens at other wavelengths?

f. Why should Prussian blue show the same pattern in opto-electrochemical response to H₂SO₄ and why should there be no dependence on scan rate/ionic strength? There is what I would have thought a clear relationship between refractive index of the solution and concentration (*ACS Energy Letters* 8 (4), 1785-1792) which would influence the results – perhaps I am missing the authors arguments a bit.

2. I find it difficult to understand why the authors suggest it is only solution resistances that contribute to/explain their observations. There has been quite some work using SECCM by the Unwin and Kanan group (*Nature Materials* 20, 1000–1006 (2021)) showing on even seemingly homogeneous Au substrates there is a large degree of heterogeneity leading to variation in reaction profiles. Why can this be ignored here? I realise this is the crux of their paper but even after several re-reads I am lost.

3. The highlight of the work is the electrochemical impedance microscopy, an outstanding technique previously developed by the same group (*Nature Communications*, 2316 (2022)). But it is based on optical data as I mention above that I do not feel can yet be seen to be reliable yet.

4. From a technique viewpoint I do not find the work is conceptually taking us further given the optoelectrochemical methods have already been published by the group and the widefield optical techniques are common-place. Indeed, even edge effects in electrocatalysis are quite well studied e.g., *J. Electrochem. Soc.* 169 096519. (Though I agree that considering the behavior in the solution is a novel question).

5. I do not find it a strong argument to say that because the same impedance maps are observed for glassy carbon as for Au that the results are proven. There could be the same systematic uncertainties underlying measurement of the two and in any case the text here gives little physical intuition for why it would be solution resistances that give rise to the observations. Is there no way they can either model the solution or perform some experiments on it e.g., using fluorescently labelled voltage dyes to get a more direct picture of what the solution is doing.

6. While a different system the modelling seems similar to that done in some ways for the tip in SECCM *Anal. Chem.* 2017, 89, 13, 7273–7276 is there some implications from the work of this paper for that field?

Minor comment

1. Generally, while the figures are nice, I find the manuscript not well written. This is not the job of peer review to fix per se but it makes the work extremely difficult for me to follow e.g. 'we discover that the electron transfer is always significantly sooner (by milliseconds), rather than faster in essence, at the periphery region than the central area..' – what does 'rather than faster' mean here? There are a numerous phrases that are poorly constructed which I struggled to follow throughout.
2. There are very few error bars on the data – when you are considering 2% changes in reflectivity it would be good to know what the error is (I acknowledge the authors discuss noise in their SI).

In summary I think the authors ask a nice question about the role heterogeneities in solution resistance play on electrode heterogeneities, but I find the approach taken falls far short of answering the problem. Most of the results rely on modelling and taking correlation as causation. While I understand my comments may seem very harsh I am genuinely trying to improve the reliability of the work such that results from opto-electrochemistry can be built on.

Reviewer #4 (Remarks to the Author):

The communication reports deployment of reflective optical microscopy for in-situ observation of Au and Prussian blue redox reaction on electrode so that the "local" reaction intensity, as judged by change in refractive index, can be observed. Using this technique, the authors first collected the cyclic voltammogram of the Au and found a good agreement between the "global" optical current and the "global" electrochemical current. This experiment offers convincing evidence for the validity of "global" optical current.

Using the local optical current, the authors found the difference in optical current at the disk and edge area of the electrode, and the edge area reached the reductive peak earlier than the centre area, as shown in Figure 2b in the manuscript. The authors postulated the shift in peak potential to possible solution phase charge transport instead of edge effect caused by radial diffusion by citing a literature with different experimental conditions (methylviologen vs. Au). As suggested by Supplementary Figure 2, the peak shift increased with increasing scan rate, but the time difference

remained around 25 milliseconds, which according to the authors, ruled out the influence of diffusion effect.

The authors then mapped the charging time constant for the non-Faradaic process of the electrode which ranged from 8 to 12 ms as shown in Fig. 3c, suggesting that the nonuniform charge transport rate in the solution side of the interface. The claim was further supported by the EIS data where the edge showed less resistance at high frequencies, which also agreed with FE simulation of the potential distribution.

At last, as evidenced by retracting the Au electrode in experiment and via simulation, the radial potential difference and the edge effect was removed.

In summary, after carefully reviewing the manuscript and the SI, I found the manuscript concise, well-written and easy to understand. However, there may need additional evidence regarding how the authors ruled out the diffusion effect and attributed the potential edge effect to nonuniform solution resistance. The manuscript may be ultimately publishable with additional evidence. There are also some minor suggestions to help the authors improve the manuscript.

Suggestions about additional evidence to support the author's conclusion.

1. The authors' discussion of the potential shift edge effect can be divided into two scenarios, Faradaic and non-Faradic. I am personally convinced with the discussion with the non-Faradic scenarios and more curious with the Faradic scenario. In the Mapping the Distribution of Apparent Electron Transfer Kinetic section, the author ruled out the possibility of diffusional edge effect by citing a literature with a different experimental setup and empirically determined that for an electrode with a 1.5 mm radius, there should be no edge effect.

Although 1.5 mm radius electrode is typically considered a macro electrode as the edge effect caused by radial diffusion was dominated by linear diffusion current, but considering it is not significantly different from the ~100 micron size threshold as mentioned the authors in page 6, the diffusional edge effect should still be carefully evaluated in this scenario. It's totally possible for radial diffusion to increase the current density by a few percentages, which could convolute the results. Is it possible that radial diffusion also plays part of the role, or the non-uniform resistance was caused by non-uniform mass transport?

The authors may perform FE simulation to reveal the magnitude of current density and/or to perform additional experiments by increasing/decreasing the electrode size to see if diffusional mass transport really plays a role. The results of this additional simulation/experiment may greatly support the authors' conclusions.

2. In the last experiment, retracting the electrode also removes the radial diffusion, which seems to confirm the contribution of radial diffusion effect?

A few suggestions to improve the manuscript:

1. As the Prussian blue redox reaction equation is mentioned in Supplementary Fig. 3a, please also include the Au redox equation in the text. According to S. B. Brummer and A. C. Makrides 1964 J. Electrochem. Soc. 111 1122, this reaction depends on some solution phase species, which mass transport may play some role.
2. What's the resolution of the microscopy? For example, when observing the 3mm diameter gold electrode, how many pixels are on the diameter line? Maybe mention this in the relevant section to help readers understand the resolution of the microscope and thus the spatial resolution of "optical current".
3. On page 5, line 146, there is a typo. The variation of reflectivity initiates (instead of "initial") and completes earlier than.....

Responses to Reviewers' Comments

Reviewer #1 (Remarks to the Author):

In this manuscript, Dr. Wang and colleagues reveal that electron transfer occurs significantly faster at the periphery of the electrode than in the central region during redox reactions, as observed through optical microscopy. By integrating multiple electrochemical techniques with electrochemical impedance microscopy and COMSOL simulations, this study spatially correlates this variation to the non-uniform distribution of solution resistance across the electrode surface. This is a very interesting phenomenon that could be significant for electrochemical field which could impact the catalyst design research and industry. Given the importance of this paper, I strongly recommend to publish the paper in Nature Communications with minor revisions.

Response: We appreciate much the strong recommendation and very helpful suggestions from the reviewer.

1. In this manuscript, the authors provide the electrolyte as the 0.01M H₂SO₄ solution. Could the authors explain the reasons for this option? Will the spatial difference still be observed on the electrode surface during the redox reaction if the concentration increases to 1 mol/L?

Response: The presented electrolyte concentration of 0.01 M is selected primarily because in this case the peak potential differences (and the charging time constant distribution) are prominent and their presence can be readily revealed. According to this reviewer's comment, we also supplemented the results for the cases with higher concentrations of sulphuric acids between 0.01 M and 0.5 M at 2 V s⁻¹ which all show clear peak potential differences along the radial direction (Figure R1). This higher scan rate applied, as compared to the 100 mV s⁻¹ used in the main text Figure 1, is necessary as the establishment of the interfacial potential would occur much faster at the very high electrolyte concentration.

Figure R1. Gold reduction peak potential differences (with that of the electrode centre) along the electrode diameter in 0.01 M to 0.5 M H_2SO_4 at 2 V s^{-1} .

2. In Fig. 1b, the authors choose the different points and share the optical image during the CV scanning process. Specifically, we are interested in knowing why at some points such as E2 and E5 are selected as inflection points while E3 is at 1.5 V, which is not the inflection point of reflectivity. Additionally, we would like to know if the delays between the potential and reflectivity, like E3, are common in the data and why they occur.

Response: We clarify that the selection of the points E_1 - E_6 is based on the characteristic points in the optical CV, namely the differential of the intensity curve, rather than on the intensity variation itself (Figure R2a). To make it clearer, we have added the points in Figure R2b. Specifically, at the points E_2 (0.95 V) and E_4 (1.05 V) the current just witnesses faradaic rises due to the gold oxidation and its oxide reduction respectively, whereas E_3 (1.50 V) and E_5 (0.70 V) correspond to the near ends of the two redox processes respectively. The noted delays are due to the unfinished redox reactions – please note that the point E_3 is right at the reverse potential 1.5 V of the optical CV but the gold oxidation is not fully finished such that the reflectivity further keeps decreasing a bit more. We apologise for confusion caused by the mislabelling in the original Figure 1b which has now been corrected.

Figure R2. (a) Optical images and (b) mean reflectivity (upper plot) of the overall electrode surface as a function of applied potential (bottom plot). (c) Cyclic voltammetry of the gold electrode (blue) overlaid with derived optical current (orange). Scan rate 100 mV s^{-1} .

3. Even though the authors provide the COMSOL simulation results of the electricity field distribution as the working electrode (WE) did not face the counter electrode (CE), as the practical experiments knowledge the WE facing the CE could receive the best electricity field. In this case, could the authors provide some simulation results with the WE and CE face-to-face each other?

Response: We thank very much the reviewer for this inspiring discussion. We have conducted and provided above the simulation of face-to-face WE-CE configuration and the results indicate that the configuration may actually not affect the electric field evolution (Figure R3). Rather, the electrode gap is the crucial factor. A smaller gap (especially 0.5 mm) would significantly reduce the spatial differences of solution resistances.

Figure R3. (a) COMSOL simulation of electric field lines between the face-to-face WE and CE and a closer look at the profile of potential distribution from the nearby solution to the working electrode surface with WE-CE gaps of (b) 9 mm, (c) 5 mm and (d) 0.5 mm. Top view of the simulated potential distribution (converted to solution resistance) over the entire electrode for WE-CE gaps of (e) 9 mm and 0.5 mm. (g) Dependence of the radial distribution of solution resistances on the electrode gap.

4. Can you explain the reason for this uneven resistance distribution in more detail? In Fig. 5 and 6, the author seems to have changed the distribution of the solution potential on the surface through physical means. In this case, will the electric field distribution be the same as the electrode is totally in contact with the solution? Authors could artificially establish an insulating boundary along the diameter of the gold electrode surface and verify if there is a difference in response only at the edge and other areas. In addition, we are also curious about the difference in solution impedance under this uneven distribution. Is there any experimental method to quantify this difference?

Response: The reason of uneven solution resistance is inferred from both physical and chemical aspects based on our supplemented simulation results below. First, the present WE-CE configuration is simulated to exhibit an increasing electric field strength from the electrode centre to the periphery (as seen from the Figure R4a-c). A stronger local electric field thus confers ions with higher mobility (μ) for establishing the interfacial potential (Figure R4d), leading to the observation of faster charging (τ) at locations closer to the electrode periphery. The faster ion mobility directly corresponds to smaller solution resistance (R_s). We can thus understand the physics *via*

$$E \propto \mu \left(\propto \frac{1}{\tau} \right) \propto \frac{1}{R_s}.$$

Figure R4. (a) Experimentally measured top view and (b) simulated lateral view of the electric field strength profile (converted from the simulated solution potentials) in the nearby solution to the working electrode surface. (c) Microscopic depiction of electric field strength evolution from the CE (at infinity) to the WE edges. (d) Microscopic depiction of the effects of electric field strength on the local potential profile evolution in solution. EDL denotes electrode double layer.

Figure R5. (a) Photograph of the marked gold electrode. Corresponding experimentally measured (b) mapping and (c) distribution along a horizontal diameter of the solution resistances. Simulated (d) side-view solution potential evolution, (e) electric field strength evolution, (f) solution resistance mapping and (g) distribution with highlighted two separate centres and the new edges of the distribution. “Exp.” means “Experimental results” and “Sim.” means “Simulation results”.

On the other hand, the chemical understanding behind the relationship, as we infer, is that the uneven resistance distribution may intrinsically arise from the ionic crowding effects. Specifically, in response to the charge transfer in solution, as the ions move to an increasingly compact space, the electrical repulsion between the ions of the same sign would increase and leads to a larger impedance of the solution. Accordingly, the space above the central region of the electrode would witness the greatest electrical repulsion whereas the latter is alleviated at the edge as the electric field is much less intense than that in the central region. As such, the resistance distribution exhibit the spatial heterogeneity.

The electric field distribution would be very different if the electrode is totally in contact with solutions – the outer sheath, due to its electrical insulating properties, does play a key role. We therefore experimentally alter the insulating surroundings of the electrode in the following and see its response then, for which we thank very much the reviewer for this brilliant suggestion.

Accordingly we have conducted the experiment by drawing along a diameter using a 0.6-mm marker pen to create a surface passivation along the diameter (Figure R5 a-c). Consequently, we do observe the original one single hump-like radial distribution of the solution resistance being taken apart to two smaller halves with a similar, symmetric shape. Both the new two maxima and two local minima of the solution resistances are evident in the distribution over the original middle area of the electrode. These are in clear contrast to the original pattern. The results are consistent with the corresponding simulation of the electric field strength distribution (Figure R5 d-g).

5. In addition, there are some minor issues that need to be corrected:

- If the figure abbreviations in Fig. 1a are not marked in the text, they need to be used in full.
- Vs in Fig. 1b should be vs.
- In line 133, Fig. 1d missed.
- In line 169, the constant of k_0 is not provided with the meaning and explanation.
- The "space" distance in Supplementary Fig. 5a and Fig. 4a seems inappropriate, please check whether there is a problem of mixing full-width and half-width.

Response: We thank the reviewer for pointing out the detailed errors and these have all been accordingly corrected.

Reviewer #2 (Remarks to the Author):

The manuscript reports an important experimental discovery of macroscale spatial heterogeneity on the electrode surface during dynamic electrochemical potential scan, for both faradaic reaction and non-Faradaic processes. Taking advantage of wide field reflective optical imaging method developed by the lab previously that can map local electrochemical current and impedance, the spatial heterogeneity is found to be caused by the solution phase resistance variations near the electrode surface, likely contributed from the ionic crowding effect. The study is well designed with proper controls to rule out other possibilities and a theoretical modeling to explain the principle of the observed phenomena. Furthermore, a simple experimental validation with inclined electrode showing that the heterogeneity can be removed completely. The conclusion is fully supported by the data presented. This finding contributes to the fundamental knowledge of electrochemistry kinetics and could have broader impact on the study and application of fast transient electrochemistry. I believe the manuscript is worth publishing in *Nature* given the potential broader impact of the findings. I only have the following minor comments:

Response: We appreciate very much the acknowledgement of the work and the potential significance from the reviewer.

Page 10, line 274, should “smaller” be “larger”? Figure 5 C and D showing the center area solution resistance is larger.

Response: Yes, it should be larger and we have now corrected.

Minor English issue:

The English of the article could be further polished. One mistake in multiple place is using “which” without putting a comma before it.

Page 3, line 78, a comma should be placed before “which”.

Page 5, line 121, “corresponds well to that at which electrochemical oxidation of gold occurs” is better be “corresponds to the gold electrochemical oxidization potential”.

Page 12, line 337, “which” should be “, which”.

Response: We thank the review for pointing out the detailed errors and these have all been accordingly revised or checked.

Reviewer #3 (Remarks to the Author):

In this work Niu et al., use opto-electrochemical methods to explore heterogeneities in reactivity at seemingly homogeneous Au microspheres. Using bright-field optical microscopy the authors infer that edges of the electrode react sooner than the periphery, due to the solution resistance (rather than the electrode itself) being inhomogeneous. They then use their results to create homogeneous solution resistances and hence remove heterogeneities. They support their results with COMSOL simulations. While the work has merit in parts, I think it is seriously under-developed and not suitable for publication in Nature Communications (or any other journal for that matter at this stage). Once the below points have been addressed, I believe it could be suitable for a specialist physical or electrochemistry chemistry journal. Below I list major and minor issues with the work:

1. There has been a huge interest in using optical methods to explore electrochemical processes due to its 'high' read-out rate (as compared to scanning probe techniques) and non-invasiveness, as the authors highlight. However, many works combining optics and electrochemistry, (often in high-impact journals) contain very poor optical controls leading to imaging artifacts being potentially interpreted as new (unreproducible) physics. These issues I fear may also plague this work, and I have the following questions:

a. How homogeneous is the illumination of the electrode? A drawback of bright field microscopies is typically across the field of view illumination is not homogeneous, this can mean imaging at edges is challenging even if data is normalised? How do the authors tackle this issue?

Response: We agree with the reviewer that unlike the typical Kohler illumination with a bright-field condenser, the illumination through the objective of the electrode is surely inhomogeneous as shown below. However, although there could be influence of initial reflectivity on the potential-dependent reflectivity variations, we would then like to demonstrate with the following three experiments that this illumination heterogeneity, which, as the reviewer suggests, might lead to systematic optical artefacts resembling our observations, is actually not a problem.

Figure R6. Images of the electrode surface with (a) the relatively even and (d) rather uneven illumination. The corresponding (b, e) initial reflectivity distribution across the electrode surface and (c, f) the experimentally measured solution resistance distributions.

On one hand, we deliberately make the illumination greatly inhomogeneous, that is, the intensity drastically varying across the horizontal direction. We then perform the same experiment, and the qualitatively same pattern is obtained again (Figure R6). Although the latter appears noisier, this indicates that the pattern discovered is not affected by the uneven illumination.

Figure R7 (a-c) Initial reflectivity distribution across the electrode surface under the retracted state and (d-e) the corresponding solution resistance distributions.

Figure R5. (a) Photograph of the marked gold electrode. Corresponding experimentally measured (b) mapping and (c) distribution along a horizontal diameter of the solution resistances. Simulated (d) side-view solution potential evolution, (e) electric field strength evolution, (f) solution resistance mapping and (g) distribution with highlighted two separate centres and the new edges of the distribution. “Exp.” means “Experimental results” and “Sim.” means “Simulation results”.

On the other hand, we further present the initial reflectivity distribution of the retracted electrode surface which also appears very heterogeneous (Figure R7), but

the results show, as discussed in the main text, the peak potential distributions are completely eliminated in this case. This demonstrates that the pattern discovered is not a consequence of the uneven illumination. We thus argue and remind that, the charging rate (a kinetic parameter that is independent on the initial intensity) is used throughout the study to avoid the possible influence of inhomogeneous illumination.

Also, we perform another experiment where the electrode surface is passivated a diameter and the charging processes are measured. As a result, the electrode shows a discrete two-maxima hump-like radial distributions of charging time constants (Figure R5 b-c above), which is found to be well consistent with the predicted electric field strength distribution (Figure R5 d-g above), rather than the initial reflectivity mapping. In this case, the initial reflectivity does not show a distribution resembling that type of variation (Figure R5a above). This confirms that the pattern is a consequence of the electric field strength distribution, which is affected little by their initial reflectivity. Based on these reproducible results under varied conditions, it is concluded that inhomogeneous illumination does not contribute to the formation of the discovered spatial heterogeneity.

b. The authors have cropped(?) their images into circularly illumination regions – potentially due to the above problem? But this then draws into question what is the edge of the electrode? How curved is the electrode they are working with? For a low NA (0.13) objective how does the depth of field of the objective influence what they are observing? Could some tilted SEM done to at least get an idea of the electrode topography?

Response: The images of the electrode are cropped but just for a better presentation of the results, and as discussed above the heterogeneity of the illumination would not be an issue.

We have carefully checked the microscale geometry of the gold electrode and accordingly provided the SEM images of both the electrode centre and edges. Figure R8 shows that under the secondary electron scattering imaging mode, which is most sensitive to the microscale heights of a sample, the electrode does not appear any visible (systematic) height differences, which is as significant as in the middle region of the Figure R8c, over the whole surface except some random scratches.

The depth of field of the low NA objective is around 16 μm , well above the reactive layer thickness of gold (~ 0.2 nm) as discussed in the main text. Therefore, the capture of the photons of interest would not be affected by the above-mentioned minor geometric features.

Figure R8. (a) Top view of a PTFE-encapsulated gold electrode and side view of the sputtered gold electrode loaded on to the SEM sampler at a tilted angle of *ca.* 15°. SEM images of (b-c) the electrode edges and (d-e) central regions.

c. The authors are imaging a complex refractive index system i.e., through water/at a metallic interface. The imaging plane of the edges could be offset from that of the centre due to imaging through the medium. I would have expected some optical modelling e.g., in Lumerical to characterise this. Such an effect will become worse when moving to higher NAs and resolution.

Response: We accordingly predict the reflectivity of the gold electrode using the suggested software. Detailed settings for the reflection of the interface through the solution includes (1) a Finite Difference Time Domain method with a boundary condition of perfectly matched layers (2) refractive indexes of gold, PTFE, and aqueous sulfuric acids from the CRC database; (3) a plane wave of 530 nm; (4) detection of the

r
e
f
l
e
c
t
i
o
n

Figure R9. (a) Reflectivity map of the constructed PTFE-encapsulated gold electrode using Lumerical. (b) Corresponding simulated radial reflectivity distribution.

d. The correlation between reflectivity and electrochemistry at the crux of the paper is poor. Factors such as how potential influences the microscope focus position (as demonstrated in the work of the Faez group e.g., Electric-double-layer-modulation microscopy Physical Review Applied 13 (4), 044065) are not considered. This is not a trivial effect that must be discussed.

Response: To resolve the effect of focal plane on the optical measurements, we deliberately move the focal plane very far (200 microns) away from the geometrical plane of the gold electrode (Figure R10). As a result, the reported spatial differences are still present. It is thus reasonable to infer that the potential-modulated double layer that can theoretically change the focal position during voltammetry are unlikely a significant aspect to affect the optical measurements.

Figure R10. Reflectivity of the gold (a) at focal plane and (b) then 200 microns away. (c) Solution resistance mapping of the latter.

In general, we think the correlation between electrochemistry and optical measurements has been carefully demonstrated because the presented optical CV is in very good agreement with the electrical CV (Figure 1c in the main text) – both are

experimentally measured data showing the consistency, despite all those factors that might need to be considered and have been discussed so far; in addition, we have also performed chronoamperometry of the electrode at potentials of 0.6 V and 0.2 V (Figure R11), and there is again excellent correlation between the charge integrated from the measured current as a function of time and the measured temporal variation of the optical reflectivity.

Figure R11. Correlation between the (a) chronoamperogram, (b) corresponding temporal change of charge transferred and (c) the measured optical reflectivity of the gold.

e. Optical imaging at interfaces is generally tricky. This is because at this point there are large refractive index differences e.g., between the solution and the electrode. This results in chromatic aberrations where light travels a different path length through the medium resulting in enhanced, but artificial, reflection contrast at edges. This was shown recently for imaging in battery materials by Pandya et al. Nature Nanotechnology 18, 1185–1194 (2023). How do the authors overcome this? Why do they chose 530 nm as a wavelength? What happens at other wavelengths?

Response: We agree that there might be different reflection contrasts at edges from that of the central region of a sample but please note that this applies to the microparticles in the mentioned paper where the edge scattering is significant in a micrometer scale. However, we use the fairly flat gold layer and presumably only one

single layer of gold reacts as discussed in the main text. It follows that the scattering at the edges of the monolayer would be negligible for the whole surface and thus impossible to extend from the edge continuously to the centre which our reported distribution features.

The selection of the 530 nm light is because the spectroscopic absorption of gold oxide is maximum at around this wavelength and thus the 530 nm light can offer optimal sensitivity of the optical measurement (Figure R12).

Figure R12. (a) Optical set-up of the spectroscopic measurements of the gold electrode and (b) corresponding spectra measured at the different potentials of interest.

f. Why should Prussian blue show the same pattern in opto-electrochemical response to H₂SO₄ and why should there be no dependence on scan rate/ionic strength? There is what I would have thought a clear relationship between refractive index of the solution and concentration (ACS Energy Letters 8 (4), 1785-1792) which would influence the results – perhaps I am missing the authors arguments a bit.

Response: The electrolyte used in the Prussian blue experiment is 10 mM KNO₃ as described in the SI. The electrochemical responses result from the K⁺ insertion/deinsertion redox electrochemistry that alters the light absorption of the material. This experiment is to show that the qualitatively same pattern is independent of the identity of the redox active material.

We clarify that we did not claim how reported phenomenon precisely looks is independent on the scan rate/ionic strength. What we emphasise is that no matter what scan rate and ionic strength is used, the spatial differences of charging time constants are always present as long as the area and the position of the working and counter electrodes are kept (Figure R3). In other words, the presence, not the characteristics, of the phenomenon is independent of the scan rate and solution ionic strengths.

2. I find it difficult to understand why the authors suggest it is only solution resistances that contribute to/explain their observations. There has been quite some work using SECCM by the Unwin and Kanan group (Nature Materials 20, 1000–1006 (2021)) showing on even seemingly homogeneous Au substrates there is a large degree of heterogeneity leading to variation in reaction profiles. Why can this be ignored here? I realise this is the crux of their paper but even after several re-reads I am lost.

Response: We clarify and emphasise that our findings in this work are very different from the often reported the spatial imaging of electrode activity. The reasons of ruling out the possibility of surface heterogeneity of the gold activity include (1) that the surface is thoroughly polished so it is guaranteed to be flat at least at the scale of millimetre, as shown in Figure R8; (2) that microscopically, we agree that there should be gold crystal boundaries, kinks or steps in our cases, but these are the kind of atomic-level heterogeneity of activity. The latter would exhibit spatially stochastic distribution that would be against our observation of “millimetre scale, continuously and smoothly varying’ heterogeneity; (3) the fact that the gold electrode is polished each time differently and placed at different rotation angles (e.g. Figure R6, 10c, 15b), and the result show very similar patterns of the charge time constant/optical amplitude distributions.

3. The highlight of the work is the electrochemical impedance microscopy, an outstanding technique previously developed by the same group (Nature Communications, 2316 (2022)). But it is based on optical data as I mention above that I do not feel can yet be seen to be reliable yet.

Response: We admit that electrochemical optical impedance microscopy is an indirect method to measure charge transfer, different from electrochemistry directly measuring the current passing through a sample. While all the methods have their pros and cons, our method offers its strength in measuring local currents with good temporal resolutions that microelectrode electrochemistry can hardly offer.

Different from the 2022 work focusing on the absolute rate of electron transfer, this work is mainly about spatial information and thus is more of quantitative measurements at a relative scale. As discussed in this response letter, we have presented *experimental evidences* based on both (1) control experiments in many ways to rule out any systematic optical artefacts and (2) the correlation between optical CV and electrical CV to show its quantification validity. Therefore, we hope that the reviewer

could please reconsider our justifications of the physical strength and validity of the optical method used in this work.

4. From a technique viewpoint I do not find the work is conceptually taking us further given the optoelectrochemical methods have already been published by the group and the widefield optical techniques are common-place. Indeed, even edge effects in electrocatalysis are quite well studied e.g., J. Electrochem. Soc. 169 096519. (Though I agree that considering the behavior in the solution is a novel question).

Response: We clarify that the point of this work lies in the discovery of the spatial distribution of charging time constant, rather than the establishment of the method, but nevertheless, also lies in the exploitation of the unique (SPEM does not possess as discussed in Introduction) methodologic strength that leads to the discovery.

We need to emphasise that what is discovered is not an 'edge effect' as discussed in Question 2. It features the continuity of the gradual changes of the charging time constant all the way along the radial directions of the electrode surface. This is completely different from any activity heterogeneity originating from the electrode itself that would exhibit either spatial stochasticity or confinement only at the edge regions for the often reported 'edge effects'.

5. I do not find it a strong argument to say that because the same impedance maps are observed for glassy carbon as for Au that the results are proven. There could be the same systematic uncertainties underlying measurement of the two and in any case the text here gives little physical intuition for why it would be solution resistances that give rise to the observations. Is there no way they can either model the solution or perform some experiments on it e.g., using fluorescently labelled voltage dyes to get a more direct picture of what the solution is doing.

Response: We agree that the presented method maps only the responses of the electrode surface and, based on this, inferring the properties of the solutions without direct measurements on the solution side. Measuring the behaviour of a solution needs addition of chromophores that may well interfere the optical measurements of gold. Hence, we now summarise the aspects that help confirm the validity of our discovered patterns: (1) The discovered spatial differences of electrode activity is not affected as discussed by the present uneven illumination. There then would not be a consequent

systematic error leading to the presented results. (2) On this basis, both the PB and glassy carbon results are strong evidence for the experimental validity of the general discovery reported in this work. (3) The gold electrode is polished each time very differently at the atomic scale but the result show very reproducibly similar patterns of the charge time constant distributions.

Figure R13. Microscopic depiction of the effects of electric field strength on the local potential profile evolution in solution. EDL denotes electrode double layer.

Why the uneven solution resistance can lead to the discovered pattern is inferred based on our simulation results and the general knowledge of electrode double layer. At locations closer to the electrode periphery, a smaller solution resistance (R_s , resulting from a stronger electric field strength) would directly lead to higher mobility of the solution phase ions (μ). For any electrochemical process, establishing the interfacial potential needs to be done first and then, consequently Figure R13 shows that the faster moving ions would cause the observation of faster electrode charging time constants (τ). This understanding thus applies to both the processes of the electrode charging between potentials of +0.2 V and +0.6 V and the gold redox reactions. We can thus understand the physics simply *via*

$$R_s \propto \frac{1}{\mu} \propto \tau.$$

6. While a different system the modelling seems similar to that done in some ways for the tip in SECCM Anal. Chem. 2017, 89, 13, 7273–7276 is there some implications from the work of this paper for that field?

Response: We are not sure if exactly useful as it depends on specific questions of interest and SECCM experimental conditions. Nevertheless, *via* this work we suggest electroanalysis

should now in general take into account the influence of the distribution of solution resistance (or electric field distribution in solutions) instead of merely the properties of the electrode side (i.e. electrode, particle and electrode-particle contact).

Possibly, the electric field lines in SECCM are highly heterogeneous around the particle of interest (please see Science, 2000, 289, 98-101), and solution resistance comes to play a role in certain ways.

Minor comment

1. Generally, while the figures are nice, I find the manuscript not well written. This is not the job of peer review to fix per se but it makes the work extremely difficult for me to follow e.g. 'we discover that the electron transfer is always significantly sooner (by milliseconds), rather than faster in essence, at the periphery region than the central area..' – what does 'rather than faster' mean here? There are a numerous phrases that are poorly constructed which I struggled to follow throughout.

Response: The word selection of 'sooner' (rather than faster) is the core of the discovery. In general, electrons transfer through the electrode-solution interface only after a sufficient electrode potential is established. In the reported case of gold reduction, the expression of 'faster electron transfer' would mislead readers to think there is spatial differences for the reduction rates at different sites of gold when the same sufficient potentials are already established for all the sites. This is totally different from the case in our work – it is the first step of interfacial potential establishment, rather than the subsequent step of redox electron transfer, that we resolve is important and we thus focus on. This is why we emphasise that the electrode potential is established sooner at locations closer to the electrode periphery and thus the electron transfer occurs sooner (which appears to be faster though).

2. There are very few error bars on the data – when you are considering 2% changes in reflectivity it would be good to know what the error is (I acknowledge the authors discuss noise in their SI).

Response: By employing background subtraction and Gaussian fitting strategies, we have achieved a large signal-to-noise ratio across the entire distribution. The reproducibility of the same experiments is high, altogether leading to a negligible standard deviation of around 1 mV at most (Figure R14).

Figure R14. The representative radial peak potential distribution with error bars.

In summary I think the authors ask a nice question about the role heterogeneities in solution resistance play on electrode heterogeneities, but I find the approach taken falls far short of answering the problem. **Most of the results rely on modelling and taking correlation as causation.** While I understand my comments may seem very harsh I am genuinely trying to improve the reliability of the work such that results from opto-electrochemistry can be built on.

Response: We thank the reviewer for the sincere comments, but indeed, we report our discovery mainly based on experiments – only Figure 5 is simulation results, other figures Figure 1-4 and 6 are basically all experimental results. The reliability of our results builds predominantly on reproducible measurements under varied conditions with careful control experiments. We are very pleased that the other three reviewers highly acknowledge the reasoning in our work, but we also appreciate this reviewer’s genuine, critical questions from the fundamental aspects of our methodology – we agree that a truly good work should always stand up to questioning. Hence, we wish our supplemented detailed explanation of the results through this response letter would significantly help this reviewer with a further evaluation on our work.

Reviewer #4 (Remarks to the Author):

The communication reports deployment of reflective optical microscopy for in-situ observation of Au and Prussian blue redox reaction on electrode so that the “local” reaction intensity, as judged by change in refractive index, can be observed. Using this technique, the authors first collected the cyclic voltammogram of the Au and found a good agreement between the “global” optical current and the “global” electrochemical

current. This experiment offers convincing evidence for the validity of “global” optical current.

Using the local optical current, the authors found the difference in optical current at the disk and edge area of the electrode, and the edge area reached the reductive peak earlier than the centre area, as shown in Figure 2b in the manuscript. The authors postulated the shift in peak potential to possible solution phase charge transport instead of edge effect caused by radial diffusion by citing a literature with different experimental conditions (methylviologen vs. Au). As suggested by Supplementary Figure 2, the peak shift increased with increasing scan rate, but the time difference remained around 25 milliseconds, which according to the authors, ruled out the influence of diffusion effect.

The authors then mapped the charging time constant for the non-Faradaic process of the electrode which ranged from 8 to 12 ms as shown in Fig. 3c, suggesting that the nonuniform charge transport rate in the solution side of the interface. The claim was further supported by the EIS data where the edge showed less resistance at high frequencies, which also agreed with FE simulation of the potential distribution. At last, as evidenced by retracting the Au electrode in experiment and via simulation, the radial potential difference and the edge effect was removed.

In summary, after carefully reviewing the manuscript and the SI, I found the manuscript concise, well-written and easy to understand. However, there may need additional evidence regarding how the authors ruled out the diffusion effect and attributed the potential edge effect to nonuniform solution resistance. The manuscript may be ultimately publishable with additional evidence. There are also some minor suggestions to help the authors improve the manuscript.

Response: We appreciate very much the acknowledgement of the work and helpful suggestions from the reviewer.

Suggestions about additional evidence to support the author’s conclusion.

1. The authors’ discussion of the potential shift edge effect can be divided into two scenarios, Faradaic and non-Faradic. I am personally convinced with the discussion with the non-Faradic scenarios and more curious with the Faradic scenario. In the Mapping the Distribution of Apparent Electron Transfer Kinetic section, the author ruled out the possibility of diffusional edge effect by citing a literature with a different

experimental setup and empirically determined that for an electrode with a 1.5 mm radius, there should be no edge effect.

Although 1.5 mm radius electrode is typically considered a macro electrode as the edge effect caused by radial diffusion was dominated by linear diffusion current, but considering it is not significantly different from the ~100 micron size threshold as mentioned the authors in page 6, the diffusional edge effect should still be carefully evaluated in this scenario. It's totally possible for radial diffusion to increase the current density by a few percentages, which could convolute the results. Is it possible that radial diffusion also plays part of the role, or the non-uniform resistance was caused by non-uniform mass transport?

The authors may perform FE simulation to reveal the magnitude of current density and/or to perform additional experiments by increasing/decreasing the electrode size to see if diffusional mass transport really plays a role. The results of this additional simulation/experiment may greatly support the authors' conclusions.

Response: We agree that the contribution of radial diffusion needs to be more carefully examined. To this end, we demonstrate the point from two aspects in the following. First, we experimentally add hydrodynamics to the solution. An injector is placed near the electrode surface and creates strong convection in the near-electrode space while the electrode is polarised and measured (Figure R15 below). In this scenario, the measurements of reflectivity are repeated under the same conditions and set-ups. Consequently, we still observe the hump-like distribution of optical impedance and thus the solution resistance. This experiment indicates that the process of diffusion does not affect solution resistance distribution.

Figure R15. (a) Experimental setup of hydrodynamic experiments and corresponding (b) mapping and (c) distribution of the measured solution resistances.

Also, we simulate the diffusion profile of protons during voltammetry at the evolution time of 1 second (Figure R16). This time is already far longer than the millisecond scale of electrode charging processes and in this case it can be seen that the diffusion profile is still confined to a lengthscale of merely 250 μm for both ends, significantly smaller than the electrode radius. Most importantly, the profile in the middle area is completely flat due to the well-known linearity of this diffusion regime. This is in contrast to the charging time constants that continuously change from one end to the other. Moreover, the other marked difference is the curvatures of the profiles at the either end where the radial diffusional profile is much steeper than the changes of solution resistances. With these results, we confirm the exclusion of any significant contribution from radial diffusion.

Figure R16. (a) Side view of the distribution of proton profiles above the electrode surface at the time of 1 s and (b) its corresponding diffusional profile of protons. (c) Side view of electric field strength distribution and (d) the corresponding radial distribution across the electrode surface.

2. In the last experiment, retracting the electrode also removes the radial diffusion, which seems to confirm the contribution of radial diffusion effect?

Response: Based on the discussions above, the radial diffusion at the electrode edge is inferred to be insignificant in the pertinent spatial and temporal scales. It thus follows that the experiments using the retracted electrode are considered to remove the near-electrode heterogeneity of electric field distribution.

We appreciate the reviewer for these discussions which make us realise that the Abstract content '(for a gold macrodisc electrode)...electron transfer is always significantly sooner...at the periphery region than the central area.' should be more precisely written as '...at the radial coordinates closer to electrode periphery than at the very centre'. This would not mislead readers to think that our discovered pattern is mainly an edge effect.

A few suggestions to improve the manuscript:

1. As the Prussian blue redox reaction equation is mentioned in Supplementary Fig. 3a, please also include the Au redox equation in the text. According to S. B. Brummer

and A. C. Makrides 1964 J. Electrochem. Soc. 111 1122, this reaction depends on some solution phase species, which mass transport may play some role.

Response: We thank the reviewer for the suggestion and have now added the redox reaction formula: $\text{AuOOH} + 3 e^- + 3 \text{H}^+ = \text{Au} + 2 \text{H}_2\text{O}$. Based on this equation, the diffusional species of interest involved in the simulation above is thus set to be proton.

2. What's the resolution of the microscopy? For example, when observing the 3mm diameter gold electrode, how many pixels are on the diameter line? Maybe mention this in the relevant section to help readers understand the resolution of the microscope and thus the spatial resolution of "optical current".

Response: We thank the reviewer for the question and have thus added the description of the pixel size (4.3 $\mu\text{m}/\text{pixel}$) for all the main figures.

3. On page 5, line 146, there is a typo. The variation of reflectivity initiates (instead of "initial") and completes earlier than.....

Response: The typo has now been corrected.

REVIEWERS' COMMENTS

Reviewer #1 (Remarks to the Author):

The authors have addressed all my concerns and I believe the paper is ready for publication.

Reviewer #2 (Remarks to the Author):

The revision has well addressed the questions from all reviewers, and I believe it is ready for publication.

Reviewer #3 (Remarks to the Author):

The authors have done a good job of responding to my concerns with regards to optical imaging artifacts and I appreciate the effort they have put in here. With regards to my concerns on novelty I defer to the very positive reports of the other referees who's research is likely more aligned. I think the manuscript should be published in Nat. Commun. as is.

Reviewer #4 (Remarks to the Author):

The authors have made a commendable effort addressing my biggest concern regarding mass transport edge effect and provided solid simulation results. I hereby suggest the publication of this paper.